# Synthetic Iris Images: A Comparative Analysis between Cartesian and Polar Representation

**DOI:** 10.3390/s24072269

**Published:** 2024-04-02

**Authors:** Adrian Kordas, Ewelina Bartuzi-Trokielewicz, Michał Ołowski, Mateusz Trokielewicz

**Affiliations:** 1Department of Biometrics, NASK—National Research Institute, 01-045 Warsaw, Polandmichal.olowski@nask.pl (M.O.); 2Institute of Control and Computation Engineering, Warsaw University of Technology, 00-665 Warsaw, Poland; mateusz.trokielewicz@pw.edu.pl

**Keywords:** biometrics, iris recognition, synthetic images, deep fakes

## Abstract

In recent years, the advancement of generative techniques, particularly generative adversarial networks (GANs), has opened new possibilities for generating synthetic biometric data from different modalities, including—among others—images of irises, fingerprints, or faces in different representations. This study presents the process of generating synthetic images of human irises, using the recent StyleGAN3 model. The novelty presented in this work consists in producing generated content in both Cartesian and polar coordinate representations, typically used in iris recognition pipelines, such as the foundational work proposed by John Daugman, but hitherto not used in generative AI experiments. The main objective of this study was to conduct a qualitative analysis of the synthetic samples and evaluate the iris texture density and suitability for meaningful feature extraction. During this study, a total of 1327 unique irises were generated, and experimental results carried out using the well-known OSIRIS open-source iris recognition software and the equivalent software, wordlcoin-openiris, newly published at the end of 2023 to prove that (1) no “identity leak” from the training set was observed, and (2) the generated irises had enough unique textural information to be successfully differentiated between both themselves and between them and real, authentic iris samples. The results of our research demonstrate the promising potential of synthetic iris data generation as a valuable tool for augmenting training datasets and improving the overall performance of iris recognition systems. By exploring the synthetic data in both Cartesian and polar representations, we aim to understand the benefits and limitations of each approach and their implications for biometric applications. The findings suggest that synthetic iris data can significantly contribute to the advancement of iris recognition technology, enhancing its accuracy and robustness in real-world scenarios by greatly augmenting the possibilities to gather large and diversified training datasets.

## 1. Introduction

The iris pattern possesses unique characteristics that have established it as a reliable biometric feature for identification and authentication systems. It is widely recognized as one of the most secure biometric modalities, offering rich personal information and exhibiting minimal textural changes throughout an individual’s lifetime.

However, in recent years, the utilization of iris images, along with other types of biometric data, has faced limitations due to national and international regulations focused on safeguarding private data. For instance, the General Data Protection Regulation (GDPR) in the EU has imposed restrictions. These regulations have led to either restricting or banning the collection of biometric datasets entirely [1], creating challenges for researchers and developers who rely on such data for their projects.

As a result, there has been a growing interest in the development of methods for generating synthetic biometric data. These methods have the potential to expand existing datasets by introducing new identities, augment the collection of individuals already present in the datasets, or even serve as an alternative to real-world biometric data. Synthetic datasets, generated through advanced generative models, offer the potential to sustain data-driven machine learning projects without infringing on privacy regulations. Moreover, they can facilitate the development of new tools for security subsystems: presentation attack detection (PAD) and protection of the biometric samples (cancellable biometrics).

As mentioned before, current synthetic biometric data generation methods are based on generative models, particularly generative adversarial networks (GANs) [2]. In the field of generative media, one of the key metrics used is the Fréchet inception distance (FID) [3]. The FID measures the similarity between two groups based on statistics derived from computer vision features. It utilizes a feature vector generated by the InceptionV3 [4] model, which is pretrained on the ImageNet [5] dataset. Traditionally, the FID has been the primary tool for quantifying the progress of generative models. However, a recent study by Kynkäänniemi et al. [6] raises concerns about the legitimacy of using the FID universally.

Most sensors purposed for iris biometric systems operate within the 700–900 nm range of the electromagnetic spectrum, which falls into the near infrared (NIR) [7]. This range is chosen because it allows for the clear capture of iris features, especially in dark eyes, reduces specular reflections and is invisible to the user, thus also reducing pupil movement. This hardware choice poses a unique challenge for generative systems tasked with creating synthetic iris images. Such systems must not only replicate the visual style and intricate patterns of the iris but also simulate the image acquisition process in this specific light spectrum. This ensures that the generated images are not only visually accurate but also suitable for analysis within the same operational parameters as those captured with real-life iris biometric systems.

Furthermore, iris recognition systems capture images of the human eye in a Cartesian representation, like traditional photography does. This image is then transformed through iris localization into a polar representation, which includes unwrapping the iris region into a dimensionless rectangle in polar space (normalization). This is to alleviate the changes in iris location and size within the original image, as well as correct for varying degrees of pupil dilation.

As the main context of our work is to create synthetic data for biometric analysis, we propose to generate a polar representation of the iris images. This allows the model to focus on iris details without generating eye details which are irrelevant from the biometric perspective, such as eyelashes, eyelids, and eyebrows and thus increasing the efficiency of the whole process, as well as the accuracy of the output iris, featurewise.

To the best of our knowledge, every work performed so far in the area of the generation of synthetic irises has focused on the generation of distinguishable iris images. In our research, we adopted a comprehensive approach to generating synthetic iris data by exploring the potential of the generative adversarial network architecture implementation StyleGAN3 [8]. This enabled us to generate images in Cartesian representation with high personal information content and to propose a novel approach for generating iris images in polar representation. The primary contribution of this study is the generation of synthetic iris data on an unparalleled scale, leveraging the most extensive dataset to date, which includes 1327 distinct identities. A significant and novel aspect of our research is the generation of iris images in polar representation—a method not previously explored, despite its widespread use in the extraction of individual characteristics within iris recognition systems. This innovation paves the way for the incorporation of synthetic irises in polar format into training datasets for iris recognition models, thus broadening the scope of application and enhancing the robustness of these models. Moreover, we developed a specialized segmentation model for polar-form iris images, furthering the practicality and effectiveness of our methodology in the field of biometric authentication.

In our research, we undertook a detailed exploration into the visual similarity between genuine and synthetic data, utilizing a method based on Siamese networks. This approach provided us with a nuanced understanding of the similarities and discrepancies inherent in synthetic biometric data. Concurrently, we critically assessed the validity of employing the Fréchet inception distance (FID) as a metric for evaluating synthetic data within the realm of biometrics, concluding that while the FID offers valuable insights, it may not be the optimal metric due to its limitations in capturing the full spectrum of biometric data fidelity.

Our contribution extends beyond the innovative data generation techniques to include a thorough quality analysis of the generated data, measuring texture density and presenting the Fréchet inception distance (FID) between authentic and synthetic data. This allowed us to evaluate the similarity and quality of the generated data. Furthermore, we tested the biometric performance of the generated iris data using the benchmark iris recognition software OSIRIS [9] and WorldCoin’s IRIS [10], empirically confirming the utility of synthetic data in biometric applications.

## 2. State of the Art in Synthetic Iris Data Generation

Several studies indicate that the research focused on generating synthetic iris data is not as extensively explored as in other biometric modalities [1,11]. One of the earliest work in this field was conducted by Cui et al., using principal component analysis (PCA) for iris recognition and the super-resolution method for the synthesis. To create realistic iris images, some researchers have proposed solutions that utilize deep convolutional generative adversarial networks (DCGAN) [12,13]. Lee et al. used a conditional generative adversarial network (cGAN) for iris image augmentation with the main purpose of sustaining individual diversities [14]. Yadav et al. proposed another approach using a relativistic average standard generative adversarial network (RaSGAN) [15]. A cyclic image translation generative adversarial network (CIT-GAN) was presented, which relied on a multidomain-style transfer to generate synthetic images for further training PAD methods [16]. Tinsley et al. used NVIDIA’s StyleGAN3 architecture to generate VGA iris images and examined the phenomenon of identity leakage [17]. Recently, Yadav et al. presented a solution to generate synthetic iris with unique identities and method for the style code extraction from an image and proposed a novel method, iWarpGAN [18]. Furthermore, a significant contribution was made by a pioneering study, cited as [19], which presents a fully data-driven approach for the synthesis of iris images. This method uniquely allows for the variation in pupil size while ensuring the preservation of identity, addressing the challenge of accurately representing the nonlinear texture deformation of the iris. A similar objective was chosen in the work of Kakani et al. [20]. In order to preserve the identity of the generated iris, the authors used three networks to segment portions of the eye (pupil, iris, sclera), extract individual features, and finally generate synthetic images.

Our work in terms of the goal and part of the methodology has the closest relationship with [17]. There, the authors, like us, proposed to generate synthetic images of irises based on the StyleGAN3 network, but as already mentioned, in our case, the generation was carried out on the polar representation of irises, which is to the best of our knowledge, the first work conducting this approach.

Additionally, we generated synthetic data using a much larger dataset than any past efforts in this field, with images sourced from 1327 individuals. For instance, in [17], the authors used data from 26 subjects. This significant scale of data generation, combined with our use of advanced StyleGAN3 technology and gamma correction techniques, allowed us to achieve a high level of diversity and realism in the synthetic iris images we produced. This approach not only fills a crucial gap in the existing body of research by introducing polar representation but also sets a new benchmark for the scale of dataset used in generating synthetic irises, enhancing the potential for more secure and accurate biometric recognition systems.

## 3. Training Iris Image Databases

The training data for our study were sourced from a combination of publicly available iris image databases: CASIAv3 (2639 images from 395 classes) [21], a subset of ND-Iris-0405 (1283 images from 402 classes) [22], Warsaw-BioBase (1905 images from 135 classes) [23], BATH (148 images from 148 classes) [24], Biosec (1200 images from 150 classes) [25], and a proprietary dataset curated specifically for this research containing 2525 images captured in visible light, representing 97 distinct irises (only the red channel of the RGB color space was utilized). Figure 1 showcases representative data samples from each dataset.

In total, the prepared training database comprised 11,623 images representing 1327 distinct irises, and exhibited diversity in terms of eye color, pupil size, and measurement devices, encompassing images captured in both infrared and visible light, including red channel imagery.

Additionally, for the purpose of experiments, the eye images from the prepared dataset underwent preprocessing to obtain iris images in a polar coordinate representation. This process involved iris semantic segmentation using a custom model based on the Deeplab-v3+ [26] architecture, followed by the transformation from Cartesian to polar coordinates.

## 4. Generating Synthetic Iris Images

### 4.1. Our Approach

To generate synthetic iris samples, we utilized StyleGAN3 [8], an advanced generative model widely used for generating photorealistic images. During the model training process, we employed a dataset consisting of iris images scaled to a size of 512 × 512 pixels. In training the StyleGAN3 model, the gamma parameter, a regularization coefficient, plays a crucial role in determining the balance between image fidelity and diversity. It affects the weighting of different features during image generation. To optimize the training process, adjustments and a gradient-based optimization of the gamma value were implemented. Observational data revealed that excessively high gamma values led to iris textures with blurring, while overly low gamma values produced samples with limited variation and reduced realism. Addressing the difficulty in fine-tuning a single γ value for optimal results, we introduced the “gamma gradient” approach. This method involves a systematic adjustment strategy for the γ parameter, incorporating parameters such as gamma-min, gamma-max, and step increments, enabling a more precise and flexible optimization of γ to improve the balance between image quality and diversity. This method adjusts gamma—a key to managing the trade-off between detail retention and variability in generated images—incrementally at specified kimg intervals during training. Once gamma reaches the predefined upper limit (gamma-max), it resets to the lower boundary (gamma-min). This strategy aims to optimize the balance between the diversity and authenticity of the generated samples, leveraging gamma’s role in enhancing the generative model’s performance.

After achieving satisfactory visual quality of the synthesized samples, supported by observing the Fréchet inception distance (FID) between the training and synthetic images, the training process was concluded. We obtained FID values of 18.37 for VGA images and 8.75 for images in the polar space. The model states and metrics were logged every 5000 training iterations, resulting in a total of 5000 states, known as snapshots. Based on the generated snapshots, we selected the ones with the lowest FID value and others varying approximately twice from the minimum value for visual assessment. For these selected states, we generated a total of 20,000 synthetic data samples of each type, which were then resized to 640 × 480 pixels for a Cartesian representation of eye images and to 512 × 64 pixels for iris images in the polar coordinate format.

### 4.2. Synthetic Iris Images in Cartesian Representation

In iris biometrics, iris image requirements, including those concerning resolution, are defined in appropriate international standards, most importantly in ISO/IEC 39794-6:2021 [27] and its earlier revisions. The process of generation an iris using our proposed method is presented in Figure 2. Despite the FID score ranging from 13 to 15 for snapshots from 1000 snapshots, the images varied in quality. Therefore, a checkpoint was selected by the authors based on the visual assessment of the content and quality for further experiments. For the selected state, we generated 20,000 synthetic data samples, which were then resized to the default VGA dimensions of biometric samples.

For the biometric evaluation, the synthetic data underwent a semantic segmentation of the iris using a model developed specifically for this purpose, and then transformed into the polar coordinate format.

### 4.3. Bypassing Preprocessing Steps: Synthetic Iris Images in Polar Representation

A traditional approach in biometric systems includes converting the iris image from VGA to polar format at the preprocessing stage. As our research showed, generated irises in polar form showed a higher potential than their circular counterparts. This led to the proposal to skip one preprocessing step, thus obtaining better-quality synthetic samples. Given that the main focus of this work was to generate synthetic iris data for the development of biometric systems, the generation of visually distinguishable eyes becomes irrelevant. Our further approach involved creating polar representations of irises, enabling further biometric analysis. By bypassing the generation of the entire eye, our model could focus on the finer details of the iris, which may lead to improved overall algorithm performance. For this purpose, we utilized training data transformed into a polar format. The results of each step of the generative model training are presented in Figure 3. For the biometric evaluation, we developed a semantic segmentation model to remove occlusions present in the polar image. Subsequently, the synthetic data underwent masking to enable the extraction of biometric patterns.

### 4.4. Visual Similarity

One approach to visually compare polar iris images was to design and train Siamese networks [28] based on the EfficientNet [29] model. We distinguished between three types of input images in the learning process: anchor, positive, and negative images, which were provided with every iteration. The first two images always belonged to the same class, and the negative sample was randomly selected from the other classes. Two iris images were given during inference. From these, embeddings were calculated, which were used to calculate the distance using the L2 norm, also known as the Euclidean norm function. Figure 4 shows the distribution of the score symbolizing the visual similarity between the images. Based on the developed visual comparator, a comparative analysis was performed between real and synthetic iris data. We can observe that the model adapted to the data by generalization, generating irises that shared many features with their original counterparts.

## 5. Quality Assessment of Synthetic Data

### 5.1. FID Assessment for Authentic vs. Synthetic Image Quality

The Fréchet inception distance (FID) method [3], tailored for quantifying the discrepancy between real and synthetic samples, was employed for the image assessment. This metric is crucial for evaluating the fidelity of images generated by the network. For this evaluation, 25 distinct snapshots captured at various stages of the generative network’s training were selected. Subsequently, from each snapshot, a batch of 2000 images was generated using a random seed. The diversity of the training set, encompassing multiple datasets with varying acquisition and technical conditions, necessitated a comprehensive evaluation strategy. This strategy entailed conducting comparisons across three distinct sets:Authentic vs. authentic—in order to define “what is true” and to take an input threshold for subsequent comparisons of metrics, the differences of the set of true images were calculated. For this purpose, two sets of 2000 images were extracted. The images were randomly selected, ensuring that there were no duplicates in the two sets being compared. The resulting FID score was 11.82.Authentic vs. synthetic—the next step of the evaluation was to find out the differences in a set consisting only of generated images. The experiment consisted of dividing each set of generated images within each checkpoint and comparing them with each other. For the snapshots from the beginning of training (200 kimg), the FID metric indicated 15.79 for the set samples. Thereafter, it remained at the value of 19 until the last snapshot (23,200 kimg).Synthetic vs. synthetic—after learning the variations in the FID metric for subcollections from the same source (authentic, synthetic), the generated images were compared with the original ones. For this purpose, a calculation of the distance of the generated images with the true ones was performed for each of the 25 snapshots.

### 5.2. Image Sharpness

The generated data were evaluated for data quality, taking into consideration image sharpness (Figure 5). Sharpness refers to the clarity or level of detail present in an image. It is a measure of the magnitude of image gradients, which represent the rate of intensity change across neighboring pixels.

The sharpness of an image can be quantified by calculating the average value of image gradients, where higher values indicate a more pronounced transition between adjacent pixels, resulting in a sharper image with clearer edges and finer details. To address the issue of distinguishing impaired and blurred edges in iris images in polar space, which might be lost using traditional methods, the phase stretch transform (PST) was utilized [30]. This algorithm, inspired by opto-physics, can enable a robust, more consistent edge detection and preservation in the polar representation of the iris image. Sharp transitions, edges, corners are characterized by higher frequencies passing through a nonlinear frequency-dependent phase filter; based on this, the PST extracts edge information, and these edges are further enhanced by morphological operations. The implementation of this method was performed using the PhyCV Python library [31]. The PST function takes the following parameters:S—phase strength, a parameter that influences how strongly the PST emphasizes structures in the phase domain of the image. Too small a value of the parameter (S) can lead to low sensitivity to certain image features, while too high a parameter can lead to too much sensitivity in the details, which can result in a noisy image;W—warp strength, which controls the degree of curvature of the image during transformation to the phase domain, thus introducing flexibility in the transformation. Too small a value can lead to a poor adaptation of the PST to image structures and features, causing instability in the transformation results. On the other hand, too large a value of the parameter transformation (W) can lead to too much image distortion, and the structures and details in the image may no longer reflect their true appearance characteristics, resulting in a loss of information and making it impossible to interpret the results reliably.

When applying transformations to an image matrix, it passes through a Gaussian low-pass filter, whose characteristics are described by the $\sigma_{LPF}$ parameter. The denoised image is then subjected to a morphological transformation using the morph_flag parameters, and the thresholds for morphological operations are set using the $thresh_{min}$ and $thresh_{max}$ parameters. The morphological operation consists in transforming the analog features into digital features, first performing downsampling and then calculating the quantiles defined in terms of the given thresholds to finally return a binary edge image. The image is thus not analyzed using pixel brightness but the phase changes of the image instead, which contain information about its structure.

### 5.3. Texture Density (Richness)

Texture density (Figure 6), in this context, refers to the measure of detected edges relative to the total area of the masked iris image in the polar space. It is calculated by counting the number of pixels representing edges within the masked area and then dividing this count by the total number of pixels in the region. The resulting ratio indicates the number of edges that occur per unit area, allowing an assessment of the complexity and pattern of the texture in the analyzed image region. A higher texture density indicates a greater number of edges, which may be associated with increased diversity and intricacy of the texture pattern in the examined area.

### 5.4. Discussion

Analyzing the results, we observed a higher sharpness index for the generated data in polar format 30.39 (±9.98) compared to the index obtained for the real data. The sharpness index calculated for the VGA format data was close to that of the real data 15.47 (±5.97) and amounted to 15.60 (±5.59). The texture density in each analyzed case was similar and amounted to 0.68 (±0.03), 0.69 (±0.02), and 0.69 (±0.02), for polar, VGA, and real samples, respectively.

## 6. Biometric Performance Evaluation on Synthetic Iris Images

### 6.1. Proposed Methodology

In our study, we utilized two approaches for biometric performance evaluation. The first approach extensively evaluated the performance of the OSIRIS [9] feature extractor and matcher, which is based on the original concept by Daugman. It encodes iris features through a phase quantization of Gabor filtering outcomes, followed by a binary code comparison using the XOR operation to compute the normalized Hamming distance. Values close to zero indicate data from the same iris, while typical results for comparisons of different irises should be around 0.5. Our analysis focused on synthetic iris images to investigate the extent to which these data carried personal information and to examine the potential risk of identity leak, if there was one. We conducted a biometric performance evaluation using both real data described in Section 2, and synthetic data in two analyzed ISO-compliant formats: Cartesian and polar space.
Cartesian representation: The first experiment utilized synthetic data generated in the VGA format. Eye images underwent comprehensive processing, including iris segmentation on VGA images using a dedicated iris segmentation model, followed by transformation to the polar domain and feature extraction using OSIRIS.Polar representation: The second experiment involved synthetic iris images generated in the polar domain. The generated images underwent semantic segmentation using a custom model and were subsequently encoded using the OSIRIS feature extractor. 
For the comparative analysis of biometric systems’ performance on generated data, we employed the WorldCoin open-iris system [10]. This approach allowed us to directly utilize our previously generated iris data in polar coordinates, integrating seamlessly with the WorldCoin pipeline at the feature extraction stage. The Gabor filters [32], employed for the feature extraction, facilitated the capture of distinctive iris patterns, which were then encoded into compact, mathematical representations. The subsequent matching process, leveraging the masked fractional Hamming Distance, enabled us to quantitatively assess the similarity between iris codes, thus evaluating the effectiveness and accuracy of the WorldCoin system in processing and recognizing synthetically generated iris data. This methodology provided a comprehensive framework for benchmarking the WorldCoin system against traditional biometric recognition systems.

The biometric analysis included comparisons of:Within-class real image matching (authentic genuine pairs),Between-class real image matching from different irises (authentic impostors),Matching between authentic and synthetic data (authentic to synthetic samples),Matching between synthetic data (synthetic to synthetic samples).

### 6.2. Results for the Biometric Evaluation

In our analysis, we observed that approximately 1.2% of the VGA synthetic data displayed visual inaccuracies with noticeable artefacts, while only 0.4% of them exhibited defects specifically in the iris region (Figure 7). For synthetic polar data, artifacts accounted for only 0.05% of the data.

The biometric comparison results for data in polar and Cartesian representations were similar, with their mean and standard deviation of 0.4705(±0.0212) and 0.4685(±0.0203), respectively. For comparisons between authentic and synthetic samples, the results were 0.4685(±0.0202) and 0.4736(±0.0194), respectively, for the polar and Cartesian space. The distribution of the comparison results in the form of boxplots are presented in Figure 8. In the case of VGA iris images, we can observe several comparisons below a Hamming distance of 0.3, indicating a similar individual content between the samples. This may suggest a higher likelihood of identity leak in the case of VGA images compared to polar images.

For the OSIRIS system, the comparison results obtained at the false acceptance rate of 0.00% revealed a false rejection rate of 0.380% in polar space, and 0.611% for VGA images. The observed EER was 0.0148% for authentic data, demonstrating a high level of accuracy in identifying authentic iris patterns. Interestingly, the EER further decreased to 0.0003% for the comparisons between authentic and synthetic data in the polar coordinate system, highlighting the system’s exceptional ability to distinguish between real and artificially generated iris patterns in this format. Conversely, for data represented in the Cartesian coordinate system, the EER increased to 0.0312%, indicating a relative decrease in matching accuracy. It is important to highlight that without the incorporation of the gradient adaptation method—“gamma gradient”—the synthetic samples exhibited issues such as being blurred or unnaturally textured. Consequently, more than 30% of these samples failed to pass the quality assessment process. When the quality assessment process was bypassed, the EER significantly increased, exceeding 10%.

For the WorldCoin approach, the EER for real data was found to be 0.0221%, while comparisons between authentic and synthetic data for polar representation yielded an EER of 0.0029%.

This comparison highlights the nuanced differences in performance and reliability between the OSIRIS and WorldCoin systems when dealing with both genuine and synthetic iris data, demonstrating the effectiveness of the “gamma gradient” method in enhancing the quality of synthetic samples for biometric verification processes. Results of the biometric comparison for each type of pairs are presented in Figure 9. It has been noted that the distributions for genuine samples are quite broad for OSIRIS, with a mean and deviation of 0.1960 (±0.0604), while for WorldCoin, it was 0.2155 (±0.0635). In the case of impostors from authentic data, the distributions were, respectively, 0.4430 (±0.0289) for OSIRIS and 0.4625 (±0.0335) for WorldCoin. The distributions for the comparisons between real and synthetic data were close to interclass comparisons and were characterized by a smaller deviation, with OSIRIS and WorldCoin at 0.3736 (±0.0202) and 0.4530 (±0.0207), respectively.

Figure 10 illustrates a visual comparison of several exemplary iris samples: real versus synthetic, with a focus on textural similarity. The selected samples were those that appeared visually similar when examined side-by-side. However, despite their visual resemblance in texture, these pairs did not show any close similarity in the biometric feature space, as indicated by their Hamming distances (HDs). None of the compared pairs achieved an HD lower than the 0.3 threshold, which is crucial for reliable biometric identification. This discrepancy suggests that while synthetic iris patterns can be visually convincing, they do not necessarily replicate the unique biometric features needed for accurate identity verification, highlighting a current limitation in the synthetic generation of biometrically precise iris images.

## 7. Conclusions

In this study, we conducted the training process of generative models to create photorealistic iris images based on a training dataset of 1327 different and diverse irises. Utilizing the StyleGAN3 model, we successfully generated diverse irises in both Cartesian and polar coordinate formats, achieving high individual variability in the generated data. Despite retaining visual similarity to the training data and exhibiting high-quality imagery, the synthetic data generated using a tailored model with an implemented gamma correction learning technique demonstrated a significant diversity in biometric features. Utilizing a large dataset source, no leakage of personal information was identified when compared with a reference dataset. However, in the case of data represented in Cartesian coordinates, we identified the possibility of identity leakage, calling for caution in practical applications of such data. Moreover, we tested the performance of two different iris recognition systems, OSIRIS and WorldCoin, both of which exhibited varied biometric feature differentiation in the generated data. Furthermore, the generated data and iris segmentation models operating on synthetic data (including in polar form) can support the process of training biometric recognition models for irises.

While analyzing the generated images, we encountered certain issues with the quality of the Cartesian representation of images. These images were taken from varying distances from the face, leading to distortions in some cases. However, these issues were not observed in the polar coordinate images.

Furthermore, we noticed a problem related to the calculation of the FID measure. Popular models used for the FID calculation were typically trained on the general-purpose ImageNet dataset. Therefore, it is essential to develop a specialized model based on a dedicated biometric dataset to ensure reliable measurements.

Despite retaining visual similarity to the training data and exhibiting high-quality imagery, synthetic data generated using a tailored model with an implemented gamma correction learning technique demonstrated significant diversity in biometric features. Utilizing a large dataset sourced from 1327 individuals, no leakage of personal information was identified when compared with a reference dataset. Furthermore, the generated data and iris segmentation models operating on synthetic data (including in polar form) can support the process of training biometric recognition models for irises.

In conclusion, our research in the field of synthetic iris image generation has made a significant contribution to the advancement of biometric data generation techniques, which is crucial given the need for vast, diverse training data for the development of biometric methods with privacy and respect towards the real, authentic data collected from volunteers in mind. However, exercising caution in the selection and curation of training data, along with further research in this area, is necessary to improve the quality and reliability of generated data in practical biometric applications.

## Figures and Tables

**Figure 1 sensors-24-02269-f001:**
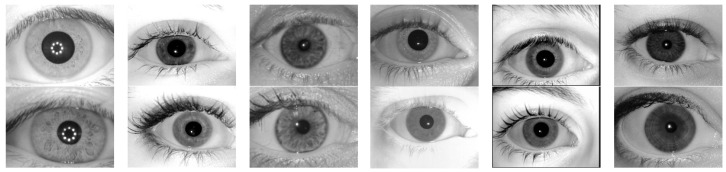
Example iris images from training datasets. From the left: CASIAv3, ND-Iris-0405, Warsaw-BioBase, BATH, Biosec, our in-house dataset.

**Figure 2 sensors-24-02269-f002:**
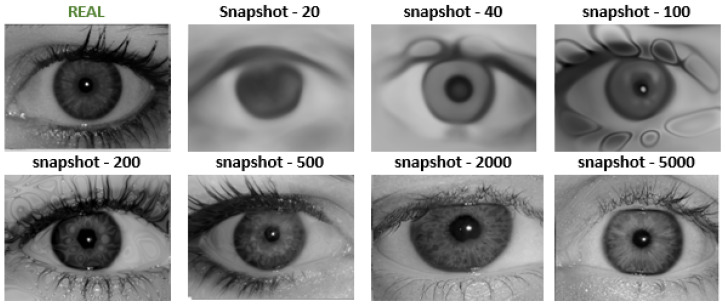
Samples presenting original training data and synthetic iris images in Cartesian representation for various training snapshots.

**Figure 3 sensors-24-02269-f003:**
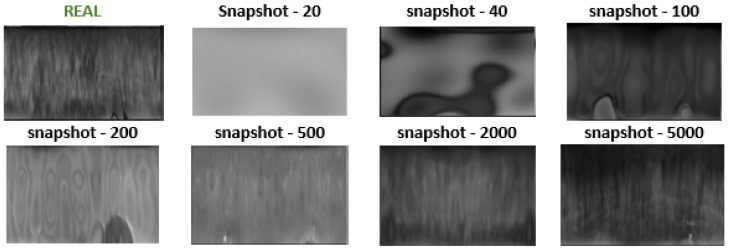
Samples presenting original training data and synthetic iris images in polar space for various training snapshots.

**Figure 4 sensors-24-02269-f004:**
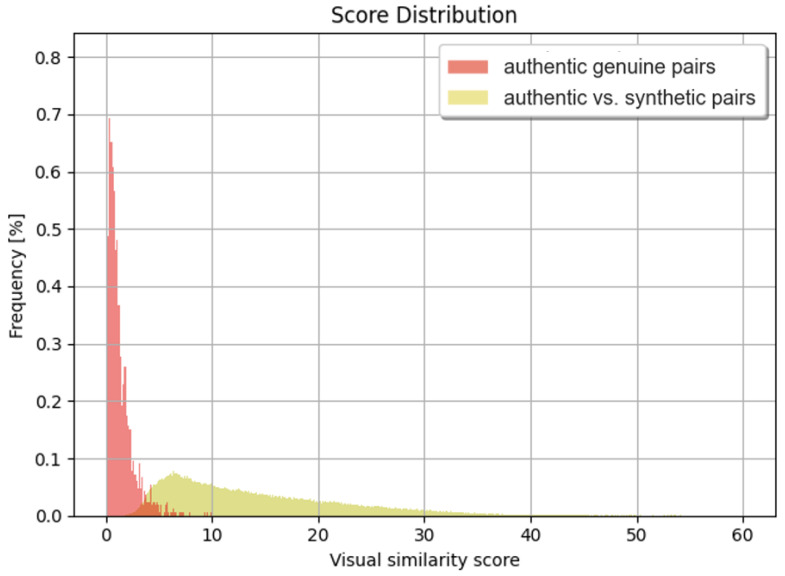
Distribution of similarity scores for genuine pairs and authentic vs. synthetic iris images.

**Figure 5 sensors-24-02269-f005:**
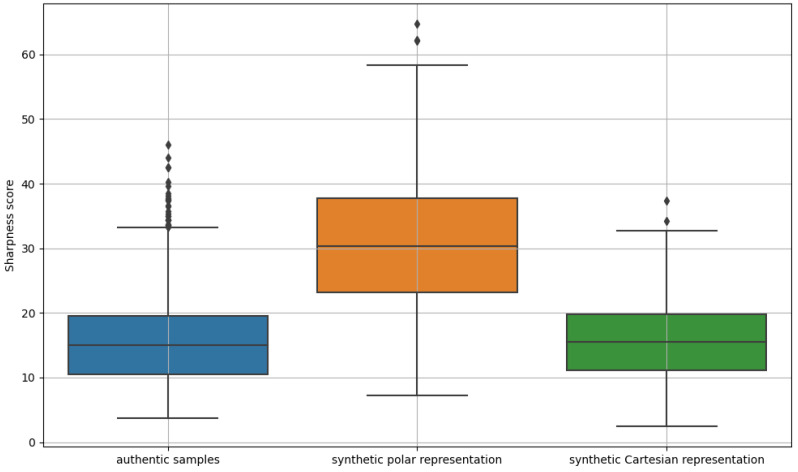
Boxplots of the sharpness indicator for real samples and synthetic data in polar and Cartesian representations, respectively.

**Figure 6 sensors-24-02269-f006:**
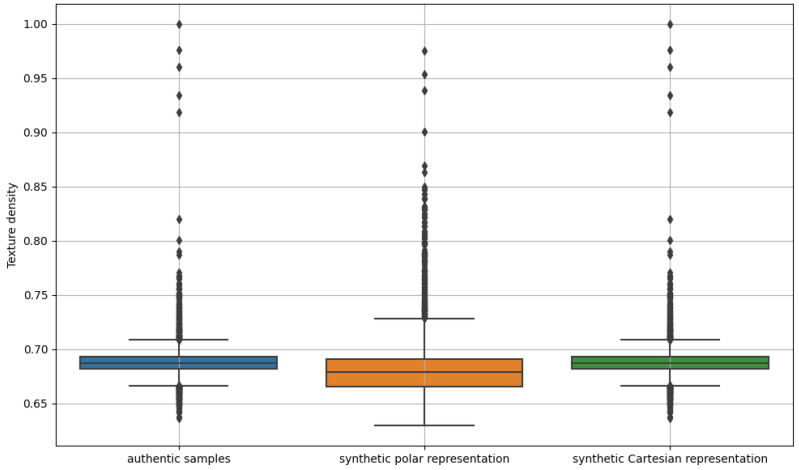
Boxplots of the PST-based texture density indicator for real samples and synthetic data in polar and Cartesian representations, respectively.

**Figure 7 sensors-24-02269-f007:**
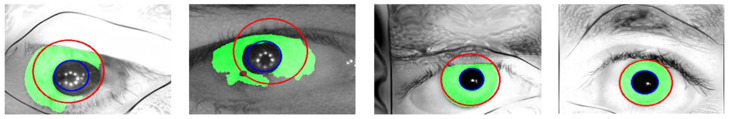
Examples of a Cartesian representation of synthetic eye images with denoted segmentation results: the first two images depict distortions in the iris region, while the last two images show undistorted irises. Red and blue circles indicate the detected iris boundaries.

**Figure 8 sensors-24-02269-f008:**
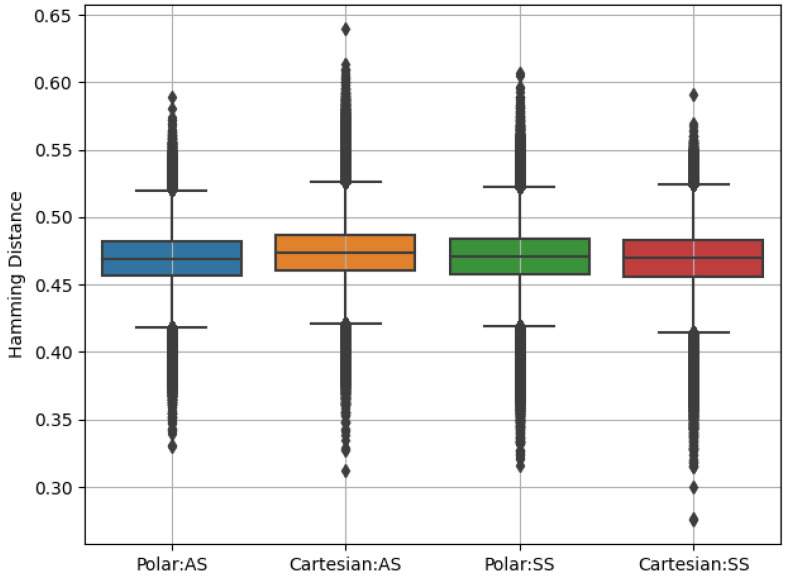
Boxplots showing the distributions of synthetic data comparison scores to compare the discriminatory power for polar and Cartesian representations. From the left: the first two boxplots show comparisons between synthetic data for polar and Cartesian spaces, and the next two boxplots correspond to comparisons between authentic samples and synthetic data for polar and Cartesian spaces.

**Figure 9 sensors-24-02269-f009:**
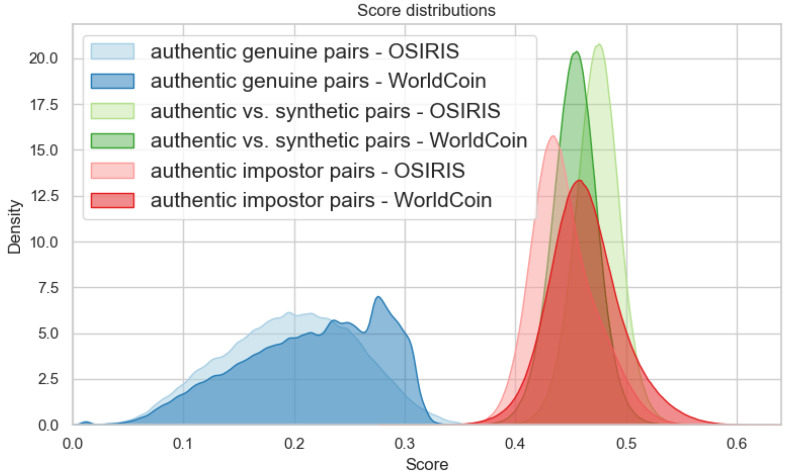
Distribution of comparison results for various comparison types within the polar representation method across two iris verification methods.

**Figure 10 sensors-24-02269-f010:**
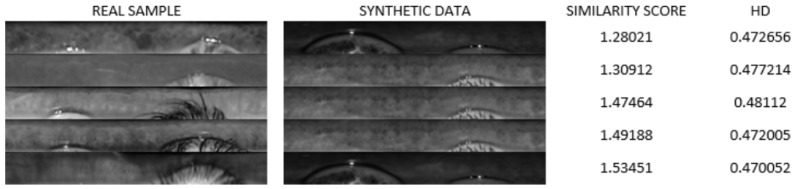
Comparison of sample data, real and synthetic, showcasing their visual similarity and Hamming distance.

## Data Availability

The dataset presented in this article is not readily available because they include iris data from our proprietary collection, which cannot be shared due to the sensitive nature of the biometric data and privacy concerns. However, we can provide synthetic data or the model for generating such data upon request, to support further scientific research within ethical and privacy guidelines. Requests to access the synthetic dataset or the generation model should be directed to deepfake@nask.pl.

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
