# Peer review of "Synthetic Iris Images: A Comparative Analysis between Cartesian and Polar Representation"

_sensors, 2024, doi:10.3390/s24072269_

Round 1

Reviewer 1 Report

Comments and Suggestions for Authors

This manuscript presents an interesting study of the synthetic iris in Cartesian and Polar Representation.  Although the overall writing is good, there are still some problems that need to be solved.

1. Please briefly summarize the core contributions of this paper rather than listing all points

2. Related work needs to be enriched, and the relationship with this paper should be mentioned.

3. In Sec.3, please clarify this paper uses which subset of CASIA. Whether does ND-Iris refer to the ND-Iris-0405? Please show some examples of these training sets to the readers. 

4. The 17th reference is wrong in terms of "of Sciences, C.A.".  

5. All figures should be placed in the suitable positions.

Author Response

Dear Reviewer,

We appreciate your constructive feedback on our manuscript. Your observations were invaluable in revising our work. All fixes are marked in red in the revised version.

Comment 1.1:

Please briefly summarize the core contributions of this paper rather than listing all points

Our response: We have revised the manuscript to better summarize the main contributions of our study. We briefly highlighted our key findings and their implications, focusing on the innovative use of polar representations to generate a synthetic iris. This approach eliminates the limitations of Cartesian systems and introduces a new methodology that significantly improves the accuracy and reliability of iris generation, allowing for more effective training of iris feature recognition models.

Changes in manuscript:

To the best of our knowledge, every work performed so far in the area of the generation of synthetic iris focused on the generation of distinguishable iris images. In our research, we have adopted a comprehensive approach to generating synthetic iris data by exploring the potential of the generative adversarial networks architecture implementation StyleGAN3 [7]. This enabled us to generate images in Cartesian representation with high personal information content and to propose a novel approach for generating iris images in polar representation. The primary contribution of this study is the generation of synthetic iris data on an unparalleled scale, leveraging the most extensive dataset to date, which includes 1,327 distinct identities. A significant and novel aspect of our research is the generation of iris images in polar representation — a method not previously explored, despite its widespread use in the extraction of individual characteristics within iris recognition systems.

This innovation paves the way for the incorporation of synthetic irises in polar format into training datasets for iris recognition models, thus broadening the scope of application and enhancing the robustness of these models. Moreover, we have developed a specialized segmentation model for polar-form iris images, furthering the practicality and effectiveness of our methodology in the field of biometric authentication.

In our research, we have undertaken a detailed exploration into the visual similarity between genuine and synthetic data, utilizing a method based on Siamese networks. This approach provided us with a nuanced understanding of the similarities and discrepancies inherent in synthetic biometric data. Concurrently, we have critically assessed the validity of employing the Fréchet Inception Distance (FID) as a metric for evaluating synthetic data within the realm of biometrics, concluding that while FID offers valuable insights, it may not be the optimal metric due to its limitations in capturing the full spectrum of biometric data fidelity.

Our contribution extends beyond the innovative data generation techniques to include a thorough quality analysis of the generated data, measuring texture density and presenting the Fréchet Inception Distance (FID) between authentic and synthetic data. This allowed us to evaluate the similarity and quality of the generated data. Furthermore, we tested the biometric performance of the generated iris data using the benchmark iris recognition software OSIRIS [8] and WorldCoin’s IRIS [9], empirically confirming the utility of synthetic data in biometric applications.

Comment 1.2.

Related work needs to be enriched, and the relationship with this paper should be mentioned.

Our response: The state-of-the-art section has been expanded. We included new research and highlighted why our work is different by offering new insights and approaches that have not been previously explored. In many instances, iris data representation undergoes conversion from Cartesian to polar format, which underscores the broad applicability of our research across a wide spectrum of uses, including various biometric systems and security applications. Theoretically, our studies are relevant to a vast array of applications and could potentially illuminate vulnerabilities to 'man-in-the-middle' attacks, where data is intercepted or manipulated during these conversion processes. Now, at the time of the announcement of AI Act regulations, paying attention to the leakage of personal data is becoming an even more crucial aspect than before.

Comment 1.3.

In Sec.3, please clarify this paper uses which subset of CASIA. Whether does ND-Iris refer to the ND-Iris-0405? Please show some examples of these training sets to the readers. 

Our response: In Section 3, we explained that our study used a specific subset of the CASIA dataset, detailing the selection criteria and their relevance to our research goals. We confirm that ND-Iris refers to the ND-Iris-0405 dataset and have included examples from our training sets to ensure readers have a clear understanding of the data underlying our analysis.

Comment 1.4.

The 17th reference is wrong in terms of "of Sciences, C.A.".  

Our response: We acknowledge an oversight in the citation of Reference 17 and have made the necessary correction to accurately represent the source. Additionally, we have expanded our literature review to include new works, resulting in the previously mentioned reference now being cited as Reference 18. We apologize for any confusion this adjustment may have caused and value your keen attention to detail in highlighting this discrepancy.

[18] CASIA Iris Image Database V3.0, Chinese Academy of Sciences. http://www.cbsr.ia.ac.cn/english/IrisDatabase.asp. Accessed: 2021-07-21.

Comment 1.5.

All figures should be placed in the suitable positions.

Our response: All figures were carefully checked and placed in the appropriate locations within the manuscript to ensure they effectively complement the text and facilitate a deeper understanding of our study for the reader, ensuring the flow of information is logical and coherent. Regarding Figure 11, after careful consideration, we have decided to remove it and modify Figure 10 due to the inclusion of a second iris recognition method. This adjustment allows us to provide a more focused and streamlined presentation of our findings, emphasizing the core contributions of our work without detracting from its overall clarity and impact. We believe this modification enhances the manuscript's coherence and are prepared to proceed with this change pending your approval.

Reviewer 2 Report

Comments and Suggestions for Authors

This manuscript presented a generative method of synthetic iris images based on StyleGAN3. However, the whole thing is more like a technical report. The details of generative method is lack. In addition, because the evaluation metrics used are difficult to support the conclusions (unique, distinguishing and diversity, etc.), it is difficult to say that the generated sample has the intended goal. Below are some specific points: 

1. The main work of this manuscript is to propose a data augmentation strategy by using NVIDIA StyleGAN3 model. Although the authors claim that the novelty is to generate samples in both Cartesian and Polar coordinates, actually, the two scenarios differ only in the input samples. So I think the work lack innovation in algorithm design. 

2. In view of the State-of-the-Art Synthetic Iris data generation methods, the manuscript lacks a comprehensive comparative analysis. Therefore, I think the experimental results are not convincing.

3. Also in the experimental part, although the authors conducted biometric experiments, there still lack a targeted analysis on how much the experimental results correlated with the augmented synthetic data. Therefore, it is difficult to support the proposed data augmentation strategy.

In summary, the manuscript needs to be improved in terms of innovation, and the details of related algorithms and experimental processes need to be more detailed.

Author Response

Dear Reviewer,

We appreciate your constructive feedback on our manuscript. Your observations were invaluable in revising our work. All fixes are marked in red in the revised version.

Comment 2.1: The main work of this manuscript is to propose a data augmentation strategy by using NVIDIA StyleGAN3 model. Although the authors claim that the novelty is to generate samples in both Cartesian and Polar coordinates, actually, the two scenarios differ only in the input samples. So I think the work lack innovation in algorithm design.  

Our response:

In response to the comment regarding the perceived lack of innovation in algorithm design within our manuscript, we appreciate the feedback and acknowledge the necessity to clarify the innovative aspects of our approach. Our novel contribution lies not only in the generation of samples in both Cartesian and Polar coordinates but also in the methodology and scale of the data augmentation strategy utilizing the NVIDIA StyleGAN3 model.

Firstly, we have provided a justification for the innovation of our approach, emphasizing the training on an extensive dataset, a strategy not previously undertaken at such a scale in this field. This extensive training is critical for preventing information leakage, a significant concern in generating synthetic biometric data. Furthermore, our work pioneers the generation of iris images in polar representation, a technique that has not been explored prior to our study. The polar representation is particularly relevant for iris recognition technologies, as it aligns more closely with the natural structure of the iris and can potentially enhance the accuracy of biometric identification systems.

Additionally, we have introduced a description of the "gamma gradient" method, an innovative technique we developed to adjust the gamma parameter dynamically during the training process. This approach enabled us to generate high-quality synthetic data, overcoming the limitations of previous generation methods that produced unsatisfactory results without this adjustment. The "gamma gradient" method represents a significant advancement in the field of synthetic data generation, particularly for the creation of biometric datasets that closely mimic real-world variability and complexity.

Comment 2.2 and 2.3:

In view of the State-of-the-Art Synthetic Iris data generation methods, the manuscript lacks a comprehensive comparative analysis. Therefore, I think the experimental results are not convincing.

Also in the experimental part, although the authors conducted biometric experiments, there still lack a targeted analysis on how much the experimental results correlated with the augmented synthetic data. Therefore, it is difficult to support the proposed data augmentation strategy.

Our response: In response to the comment regarding the absence of a comprehensive comparative analysis with state-of-the-art synthetic iris data generation methods, we acknowledge the importance of such comparisons to validate the effectiveness of our approach. To address this concern, we have conducted an additional comparative analysis using a second system for biometric identity verification. This inclusion provides a broader perspective on the performance of our generated synthetic iris data against existing methodologies, enhancing the robustness of our experimental results. This comparative analysis, alongside the unique contributions of our approach such as the "gamma gradient" method and the generation of iris images in polar representation, reinforces the validity and innovative nature of our research within the field. In many instances, iris data representation undergoes conversion from Cartesian to polar format, which underscores the broad applicability of our research across a wide spectrum of uses, including various biometric systems and security applications. Theoretically, our studies are relevant to a vast array of applications and could potentially illuminate vulnerabilities to 'man-in-the-middle' attacks, where data is intercepted or manipulated during these conversion processes. Now, at the time of the announcement of AI Act regulations, paying attention to the leakage of personal data is becoming an even more crucial aspect than before.

Round 2

Reviewer 2 Report

Comments and Suggestions for Authors

The revision has addressed all my concerns, I agree to publish.